

# An assessment of $CO_2$ and $CH_4$ emissions in a tropical river: from the Kenyir Reservoir to the estuary

Daryl Jia Jun Lee[1], Siti Farhain Mohd Ludin[1], Wei Wen Wong[2], Liyang Zhan[3] and Seng Chee Poh[1,4]

[1] Faculty of Science and Marine Environment, Universiti Malaysia Terengganu, Kuala Nerus, Terengganu, Malaysia
[2] Water Studies, School of Chemistry, Monash University, Clayton, Victoria, Australia
[3] Key Laboratory of Global Change and Marine-Atmospheric Chemistry, Third Institute of Oceanograhy, Ministry of Natural Resources, Xiamen, Fujian, China
[4] Institute of Oceanography and Environment, Universiti Malaysia Terengganu, Kuala Nerus, Terengganu, Malaysia

Corresponding author
Seng Chee Poh, poh@umt.edu.my

## ABSTRACT

This study investigates the spatial and seasonal variations in $CO_2$ and $CH_4$ emissions from the Kenyir hydropower reservoir and its downstream Terengganu River system in Malaysia. Understanding these variations is crucial for assessing whether the greenhouse gas (GHG) budget for this aquatic continuum significantly contributes to global emissions. Malaysia's distinct monsoonal climate presents a unique opportunity to explore the influence of seasonal hydrological changes on GHG emission dynamics in inland waters. Five sampling campaigns were performed at the reservoir to investigate this, involving three longitudinal surveys from the reservoir downstream along the Terengganu River, and two time-series samplings at the estuary between November 2017 and August 2019. Our findings reveal that GHG emissions from the Kenyir Reservoir are notably higher during the wet season (97 mmol $CO_2$ m$^{-2}$ d$^{-1}$ and 2 mmol $CH_4$ m$^{-2}$ d$^{-1}$) than during the dry season (54 mmol $CO_2$ m$^{-2}$ d$^{-1}$ and 0.8 mmol $CH_4$ m$^{-2}$ d$^{-1}$). This increase coincides with increased wind speed and potential surface mixing during the wet season. Despite operating since 1985, the Kenyir Reservoir's total GHG emissions remain high compared to other global reservoirs, likely due to its tropical location and high organic carbon content. Elevated GHG emissions were recorded along the Terengganu River, near the dam discharge outlets, with gradual reductions observed downstream. Despite the estuary's smaller surface area, more GHGs are emitted there than in the river. Overall, the Terengganu River catchment emits approximately 572 Gg $CO_2$-equivalent annually, with the Kenyir Reservoir accounting for the majority (94%). The river and the estuary contribute 0.5% and 5.5%, respectively. This study highlights the substantial role of tropical hydropower reservoirs and their downstream river networks in the global GHG budget, emphasizing the need for further investigation into the factors influencing GHG dynamics in tropical river systems.

## INTRODUCTION

River fragmentation through dam installation to meet the needs for food security (including water), safety (flood protection), and electricity (hydropower) has been extensively implemented in the rapidly growing region of Asia (*Barbarossa et al., 2020*). Major river basins, such as the Wujiang (*Wang et al., 2021*), Mekong (*Liu et al., 2020*), and Amazon (*Kemenes, Forsberg & Melack, 2007*), have been fragmented by dams. This fragmentation alters river hydrology upstream and downstream, resulting in riparian wetland losses and aggregates large amounts of organic matter, affecting biogeochemical cycles within the reservoir and downstream, which in turn induces greenhouse gas (GHG) emissions. Numerous studies have reported changes in the GHG emission due to dam construction (*Maavara et al., 2020*; *Wang et al., 2018*). Large portions of GHG can rapidly escape to the atmosphere due to hydrostatic drops as bottom water passes through turbine's discharge outlets. Meanwhile, the remaining carbon dioxide ($CO_2$) and methane ($CH_4$) in solutions after water passes through a dam either diffuse into the atmosphere or undergo complex biogeochemical processes downstream of the dam.

In cascading reservoir systems, such as in the upper Mekong, the potential for $CH_4$ production could significantly increase due to the continuous entrapment of sediment organic matter over successive damming events (*Liu et al., 2020*). Long periods of water retention in deep reservoirs promote the production of $CH_4$ and $CO_2$ in the bottom layer of reservoirs due to hypoxic conditions and the accumulation of organic carbon (*Müller et al., 2019*). The discharge of GHG-supersaturated and anoxic water from hydropower dams can significantly deteriorate downstream riverine systems, affecting physical, biological, and chemical water properties (*Reis et al., 2020*; *Winton, Calamita & Wehrli, 2019*). There is growing concern about GHG emissions downstream from tropical hydropower dams. Previous studies have shown that GHG emissions in tropical dammed rivers vary significantly, ranging from 100 to 1,000 mmol $CO_2$ $m^{-2}$ $d^{-1}$ and 0.5 to 290 mmol $CH_4$ $m^{-2}$ $d^{-1}$ (*Abril et al., 2005*; *Kemenes, Forsberg & Melack, 2007*; *Kemenes, Forsberg & Melack, 2016*).

Urban estuaries are often regarded as a net source for $CO_2$ and $CH_4$, contributing an approximately 0.27 Pg C $yr^{-1}$ to the atmosphere (*Laruelle et al., 2013*). The upstream regions of urban estuaries are typically supersaturated with GHG and nutrients. At the same time, adjacent oceanic waters are usually undersaturated due to lower terrestrial organic carbon input and higher biological productivity. Freshwater input from dammed rivers increases GHG levels in estuaries, but these levels generally decrease with salinity through dilution with undersaturated ocean water. However, GHG sources and sinks in estuaries are highly variable, influenced by a combination of biogeochemical processes and tidal dynamics, which change spatially and temporally across different water conditions (*Borges & Abril, 2011*; *Geyer, Chant & Houghton, 2008*; *Pfeiffer-Herbert et al., 2019*). Urbanized estuaries are modified with breakwaters and similar structures to protect harbours and shorelines. These alterations change estuarine hydrodynamics, disrupt freshwater discharge, and increase nutrient loadings, potentially causing eutrophication, anoxia, and higher GHG emissions.

In Malaysia, despite a vast renewable freshwater resource (∼580 billion m$^3$; *Food and Agriculture Organization (FAO), 2021*), few studies have examined GHG emissions from aquatic systems. This study examines GHG emissions within a modified aquatic continuum comprising the Kenyir Reservoir and the Terengganu River. Kenyir Reservoir, one of Malaysia's largest man-made water bodies, was impounded in 1985 without vegetation clearance, leaving behind organic matter that now fuels bottom water GHG production. The downstream Terengganu River, spanning 61.5 km, receives hypolimnion discharge from the reservoir and input from other urbanized tributaries, such as the Berang, Tersat, Telemong, and Nerus Rivers. At the river mouth, the Terengganu River Estuary (TRE), a salt-wedge system modified by a semi-enclosed breakwater experiences altered water mixing dynamics due to restricted freshwater and seawater exchange. This setting, influenced by its coastal location and human interventions, offers a valuable case for studying how engineered modifications affect GHG emissions and water management.

This study conducted multiple spatio-temporal surveys to map $CO_2$ and $CH_4$ emissions, identify GHG hotspots, and assess the impact of reservoir damming on emission intensity. By integrating these results with global data, we gain valuable insights into how engineered modifications affect water quality and GHG dynamics, as well as the contribution of the Kenyir Reservoir and Terengganu River to global GHG emissions.

## MATERIALS & METHODS

### Study site and sampling

Water sampling campaigns were divided into three areas: (i) Kenyir Reservoir (main reservoir only), (ii) Terengganu River (water head and intermediate section, ≈50 km length) and (iii) Terengganu River Estuary (downstream, 15 km length) (Fig. 1). Five sampling campaigns were conducted within two seasons (dry and wet) in seven sampling stations (S1–S7) in Kenyir Reservoir. The reservoir has a maximum depth of 145 m, and the maximum water level drawdown was limited to 10 m; with a minimum dam operating level at 120 m (*Rouf et al., 2010*). Both dry (March, May, July 2018) and wet (February and November 2018) seasons field sampling campaigns were determined based on local rainfall data obtained from the nearby hydrological station (Sg Gawi station, Kenyir).

Three longitudinal samplings were conducted immediately downstream of the Kenyir Dam under two contrasting river discharge conditions. 13 sampling stations (L1–L13) were chosen depending on land use, logistics and accessibility through jetty and bridges. The sampling campaigns covered ≈50 km in river distance, from the dam outlets to 15 km from the mouth of the Terengganu River. The monthly averaged-discharge rate of the Terengganu River varied from 147 to 224 m$^3$ s$^{-1}$ (data obtained from the Department of Irrigation and Drainage Malaysia), showing relatively little variation between wet and dry seasons. The river sampling campaigns were designed to capture different discharge conditions rather than focusing on seasonal differences. Specifically, the March and April 2018 sampling campaigns were carried out during the high-discharge season, with river flows exceeding 150 m$^3$ s$^{-1}$. In contrast, the April 2019 campaign was conducted during the low-discharge season when river flows were below 150 m$^3$ s$^{-1}$.

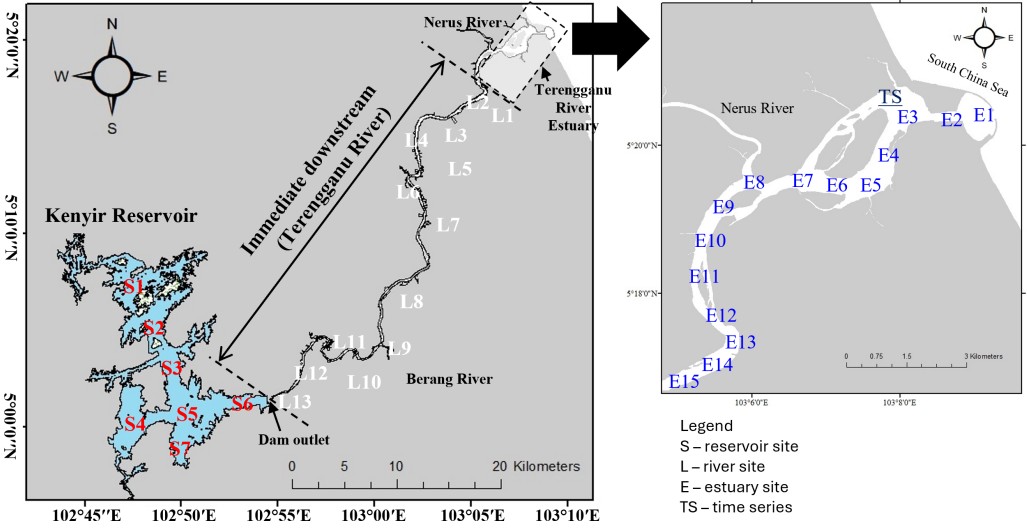

**Figure 1** Study area and sampling sites.

The estuarine sampling campaigns were divided into three longitudinal and two time series samplings. The estuarine is a shallow tropical salt-wedge estuary with a depth of 1 to 10 m (average depth: 4.7 m). The longitudinal sampling consists of 15 depth profiling stations (E1–E15) covering a transect distance of 15 km starting from the Terengganu River mouth (Fig. 1). The sampling was conducted during December 2017 (high flow), September (low flow) and December 2018 (high flow). In addition, two 24-hour stationary time series samplings were conducted at different tidal regimes in April 2019 (spring tide) and August 2019 (neap tide). For time series sampling, vertical profile physical water quality measurement was collected every hour, while GHG and nutrients sample collection for surface and bottom depths were collected every two hours.

Data S1 and Table S1 summarise the study sites and sampling activities conducted in the reservoir, river, and estuary during each campaign.

All *in situ* physical water quality measurements (pH, temperature, conductivity, dissolved oxygen, and hydrostatic pressure) were taken with a calibrated YSI 6-Series Multiparameter Water Quality Sonde (YSI 6600, USA). The vertical water column profile was done by lowering the probe until it reached a safe bottom with the assistance of a portable sonar profiler to avoid underwater equipment entanglement with submerged debris such as branches of dead trees.

## Greenhouse gases sampling and analysis

Water samples were taken manually with a 5L Niskin water sampler (General Oceanic, Miami, FL, USA) and carefully transferred *via* tubing into pre-baked 60 mL borosilicate serum bottles to ensure laminar flow and minimize bubble formation. The samples were immediately preserved with saturated mercury chloride (0.05% vol/vol), sealed with butyl rubber stoppers and crimped metal caps. The samples were transported back to the Universiti Malaysia Terengganu chemistry lab for gas chromatography (GC) analysis,

following the protocol described by *Lee et al. (2022)*. The GHG samples were headspaced by replacing one-third of the water with helium gas. The samples were then left to equilibrate at room temperature for 30 min, followed by $CO_2$ and $CH_4$ measurements using a gas chromatograph (Agilent 7980 GC system). Five mL of headspace was injected into the GC using a gas-tight syringe. The GC was equipped with 0.9 m by 1.6 cm o.d. column packed with 80/100 mesh HayeSep Q (Agilent J&W). For $CH_4$ analysis, Helium carrier gas was set at the flow rate of 21 mL min$^{-1}$. The flame ionization detector operated under the following conditions: $N_2$ makeup gas two mL min$^{-1}$, $H_2$ 48 mL min$^{-1}$, air 500 mL min$^{-1}$, and temperature 250 °C. $CO_2$ concentration was measured by a thermal conductivity detector operated at the column temperature of 120 °C and filament temperature of 200 °C. The GC column oven was operated at an initial temperature of 50 °C for 4 min, then programmed to 90 °C, and held at this temperature for 6 min. The calibration preparation procedure for GC analysis was according to *Lee et al. (2022)*.

The ebullition fluxes experiment was conducted in the shallow part of the Kenyir Reservoir. Three submersible chambers were placed about 0.5 m below the surface. 60 mL serum bottles prefilled with deionized water were used for gas collection. A tube with a 21G needle was connected from the submersible chamber nozzle to the bottom of the serum bottle (inlet), and another 21G needle near the bottleneck served as the outlet. The outlet tube's end was placed lower than the nozzle's depth.

The setup required priming to remove air from the chamber and tubing using a syringe attached to a three-way valve. Priming was complete once the setup was filled with water. The experiment ran for 24 h with samples taken in triplicate. The gas volume collected was determined by the water loss in the serum bottle (*Gao et al., 2013*); $CO_2$ and $CH_4$ concentrations were measured using GC as described above. Figure S1 shows the detailed design of the submersible chamber.

## Greenhouse gases diffusive flux across the water–air interface

The diffusive flux of $CO_2$ and $CH_4$ was obtained by estimating the dissolved GHG concentration of surface water and gas transfer coefficient, $k_{600}$ (*Ferrón et al., 2007*; *Wanninkhof, 1992*). The $pCO_2$ (or $pCH_4$) flux is given in Eq. (1).

$$Gas\ Flux = k_i K_i (pGas_{2water} - pGas_{2atm}) \tag{1}$$

where *Gas Flux* is the outgassing of $CO_2$ or $CH_4$ across air-water interface (mmol m$^{-2}$ d$^{-1}$), $k_i$ (cm h$^{-1}$) is the gas transfer velocity coefficient, $K_i$ is the solubility constant of $CO_2$ or $CH_4$ (mol L$^{-1}$ atm$^{-1}$), $pGas_{water}$ (µatm) is the partial pressure of $CO_2$ or $CH_4$ in surface water, and $pGas_{atm}$ (µatm) is the partial pressure of $CO_2$ or $CH_4$ in atmosphere.

Gas transfer velocity, $k_i$ is influenced by water temperature and salinity. Following the method specified by *Ferrón et al. (2007)*, $k_{600}$ (cm h$^{-1}$) values from each parameterizations was used to calculate $k_i$ for each gas at the recorded temperature and salinity in the field using Eq. (2):

$$k_i = k_{600}(Sc_i \div 600)^n. \tag{2}$$

For the reservoir, we used $n = -2/3$ for the wind speed <3.7 m/s and $-1/2$ for higher wind speed (*Wanninkhof, 1992*). For rivers and estuaries, where the water-air interface is

expected to be turbulent rather than smooth, $n = -1/2$ was applied (*Guérin et al., 2006; Wanninkhof, 1992*). For the Schmidt numbers, $Sc_i$ is the ratio of the kinematic viscosity of water over the diffusivity of the gas ($Sc = v/D$). The formulation of $Sc_i$ is derived from *Wanninkhof (1992)* and expressed in Eq. (3):

$$Sc = A - Bt + Ct^2 - Dt^3 \qquad (3)$$

where $t$ is in degrees Celsius (°C), and A, B, C and D are constant for the coefficients listed in *Wanninkhof (1992)* (refer to Table S2).

Since this study did not measure $k_{600}$, gas fluxes were estimated using parameterizations of $k_{600}$ reported in the literature (Table S3). The final flux determination was obtained by averaging all parameterizations applied for the reservoir and estuary study areas. In contrast, for the lake study area, the *Crusius & Wanninkhof (2003)* (C&W03) parameterization was exclusively used to estimate fluxes. C&W03 parameterization for the lake was used in the Terengganu River due to the absence of accessible water velocity and depth measurements, which are typically required for accurate $k_{600}$ estimation in river sections. C&W03, previously applied in a reservoir study, was chosen because it provides the highest transfer velocity (cm h$^{-1}$) value, offering a conservative upper limit estimate. This approach ensures a comprehensive and cautious estimation of gas fluxes within the constraints of the available data.

In this study, wind data for the $u_{10}$ (m s$^{-1}$) in $k_{600}$ was taken from *Zippenfenig (2024)* global model (Table S4) and applied in the literature $k_{600}$. The local wind speed data was extrapolated to wind speed at 10 m ($u_{10}$), according to *Crusius & Wanninkhof (2003)*, Eq. (4):

$$u_{10} = u_z \left[ 1 + \frac{(C_{d10})^{1/2}}{K} \ln\left(\frac{10}{z}\right) \right] \qquad (4)$$

where $z$ is the measured wind speed height, $C_{d10}$ is the drag coefficient at 10 m in height (0.0013; *Stauffer, 1980*), and $K$ is the Von Karman constant (0.41).

The ebullition flux was obtained by using the following Eq. (5) from *IHA (2010)*:

$$\text{Ebullition Flux (mgm}^{-2}\text{d}^{-1}) = \frac{\text{Gas Con.}(\text{mgm}^{-3}) \times \text{Gas Vol.Collected}(\text{m}^3)}{\text{Funnel Area}(\text{m}^2) \times \text{Sampling Interval}(\text{days})} \qquad (5)$$

where gas con. is the measured GHG ($CO_2$ and $CH_4$), gas vol. collected is the volume of GHG collected in the serum bottle, and funnel area is the surface area of the inverted chamber (refer to Fig. S1).

The GHG emissions were estimated using a surface area-based approach. Surface areas of the Kenyir Reservoir, the Terengganu River, and its estuary were digitized from satellite imagery. The average GHG concentration was determined from field measurements at multiple locations within each waterbody. The total GHG flux was then calculated using the Eq. (6), where the flux density was derived from *in situ* measurements and expressed in mmol m$^{-2}$ d$^{-1}$.

$$\textit{Total GHGs Emission} \left(mmol\ m^{-2}d^{-1}\right) = \textit{surface area} \left(m^2\right) \times \text{mmol}\ m^{-2}d^{-1}. \qquad (6)$$

## Data visualization and statistical analysis

The spatial distribution of GHG and fluxes was mapped using ArcGIS and the Ocean Data Viewer (ODV, AWI) software. A variable marker size approach was employed to display individual data points. Instead of simply showing coloured dots at sample points for visualizing heatmap continuity, the weighted-average gridding spatial interpolation technique available in ODV was used (*Schlitzer, 2002*). The interpolation X and Y scale-lengths were minimized to prevent overlapping coverage between data points and balance the data structure and smoothness preservation.

The dataset was characterized using descriptive statistics (minimum, maximum, average, median, and standard deviation) performed with SPSS statistical software, and seasonal trends were averaged for comparison. Since the Shapiro–Wilk test ($p < 0.05$) indicated that the data did not follow a normal distribution, non-parametric methods were used. Specifically, Spearman's rho was applied to assess correlations, and the Kruskal–Wallis test was employed to compare means across stations, depths, and periods.

# RESULTS

## Water physical properties

The reservoir was thermally stratified during dry and wet periods (Fig. S2). Surface water temperature ranged from 28.5 °C to 30.5 °C across the study sites. The average temperature difference between the surface and bottom water was significant ($p < 0.05$, ±5.0 °C). Under stratification conditions, the reservoir maintained a permanent hypoxic hypolimnion. The vertical distribution of oxygen saturation generally became hypoxic after 40 m depth, but several second oxyclines were observed between 20 and 40 m water depth at several shallow sites.

Figure S2 shows that the oxygen saturation levels (DO) differed significantly among the upstream (L1–L4), middle (L5–L8), and downstream (L9–L13) sections of the Terengganu River (Kruskal–Wallis test, $p < 0.05$). Notably, the DO level immediately downstream of the dam (L1, 1 km) was consistently lower (41 DO%) than other parts of the river (72 DO%). The spatial extent of low DO concentrations in the upstream sections (L1–L4) was directly associated with dam outflow averaged-discharge rates. During the high discharge periods in March 2018 and April 2018, with dam outflow averaged-discharge rates recorded at 163 and 154 m$^3$ s$^{-1}$, respectively, low DO levels (<50 DO%) extended 11–12 km downstream from the dam (Fig. S2). In contrast, the low flow period in April 2019, with an outflow averaged-discharge rate of 124 m$^3$ s$^{-1}$, resulted in a considerably reduced spatial extent of low DO, confining the affected area to within one km of the dam outlet.

The longitudinal and vertical profiling of salinity at the estuary indicates that the area is a partially mixed estuary, with tidal amplitudes averaging between 0.8 and 2.9 m. Figure S2 shows that the maximum seawater intrusion occurred approximately eight km from the estuary's mouth (E8–E15). The shorter saltwater intrusion observed in this study could be attributed to the presence of a breakwater outside the estuary inlet. The Terengganu estuary inlet is protected by a semi-circular coastal breakwater, which has only a small opening to the sea, thereby limiting freshwater outflows.

During September 2018 sampling (low flow), DO levels in the estuary were typically lower near the substrate and higher closer to the water's surface. In contrast, DO levels were homogeneous throughout the water column during the December 2017 sampling (high flow). The variation in DO concentrations between the two sampling campaigns may be due to seasonal differences in river discharge rates. Although river discharge data were unavailable for the December 2017 sampling campaign, the water level in December 2017 (7.8 m) was higher than in September 2018 (7.3 m). River water level is directly proportional to flow velocity (*Mohd Saupi et al., 2018*). Consequently, the estuary's dissolved oxygen levels were often not vertically stratified during high discharge events, as strong river flows tend to homogenize the water column and disrupt any oxyclines. In addition, the DO concentration in the estuary during December 2017 was much lower than during September 2018.

## GHG concentrations and fluxes

The $CO_2$ and $CH_4$ concentrations in the Kenyir Reservoir were supersaturated, with the $CO_2$ ranging from 96 to 616 $\mu M$, and $CH_4$ from 0.1 to 127 $\mu M$. Seasonally, $CO_2$ concentrations showed significant differences (Kruskal–Wallis test $p < 0.05$), while $CH_4$ concentrations did not show differences within sampling months (Kruskal–Wallis test, $p > 0.05$). There was a statistically significant difference in $CO_2$ and $CH_4$ concentrations between the sampling sites (Kruskal–Wallis test $p < 0.05$, Fig. S3).

However, $CO_2$ and $CH_4$ fluxes in the Kenyir Reservoir did not show significant spatial variations (Kruskal–Wallis test, $p > 0.05$, Fig. S3). During the dry period, $CO_2$ fluxes for the Kenyir Reservoir (S1-S7) ranged from 26 to 62 mmol m$^{-2}$ d$^{-1}$, and $CH_4$ fluxes ranged from 0.02 to 5.0 mmol m$^{-2}$ d$^{-1}$. During the wet period, $CO_2$ fluxes increased significantly compared to the dry period (Kruskal–Wallis test $p < 0.05$), ranging from 5.6 to 143 mmol m$^{-2}$ d$^{-1}$, while $CH_4$ fluxes did not show significant seasonal variation (Kruskal–Wallis test $p > 0.05$), ranging from 0.1 to 12 mmol m$^{-2}$ d$^{-1}$. Higher $CO_2$ and $CH_4$ emissions were detected in the surface water of S3, S4, S6, and S7 (Fig. 2).

Ebullition fluxes during the dry period were significantly higher (Kruskal–Wallis test, $p < 0.05$), with $CH_4$ reaching $3.79 \times 10^{-2}$ mmol m$^{-2}$ d$^{-1}$ and $CO_2$: $1.56 \times 10^{-2}$ mmol m$^{-2}$ d$^{-1}$, respectively. These rates were at least an order of magnitude higher than those observed during the wet period ($CH_4$: $3.94 \times 10^{-8}$ mmol m$^{-2}$ d$^{-1}$ and $CO_2$: $1.66 \times 10^{-4}$ mmol m$^{-2}$ d$^{-1}$).

Averaged $CH_4$ concentrations downstream of Kenyir Reservoir (Fig. 3) were significantly higher during the high discharge period ($9.1 \pm 15$ $\mu M$, range: 0.2 to 65 $\mu M$) than during the low discharge period ($1.3 \pm 1.2$ $\mu M$, range: 0.001 to 4.4 $\mu M$, Kruskal–Wallis test $p < 0.05$). High $CH_4$ and $CO_2$ concentration anomalies were observed in the upstream (L1–L4) and the middle segment (L7–L10) of the Terengganu River, indicating a potential hotspot for GHG emission.

In the Terengganu River Estuary, $CO_2$ and $CH_4$ concentrations across all sites varied between December 2017 (high flow) and September 2018 (low flow, Fig. 4). The $CH_4$ concentrations ranged from 0.2 to 20 $\mu M$ during high flow, and between 1.1 to 10 $\mu M$

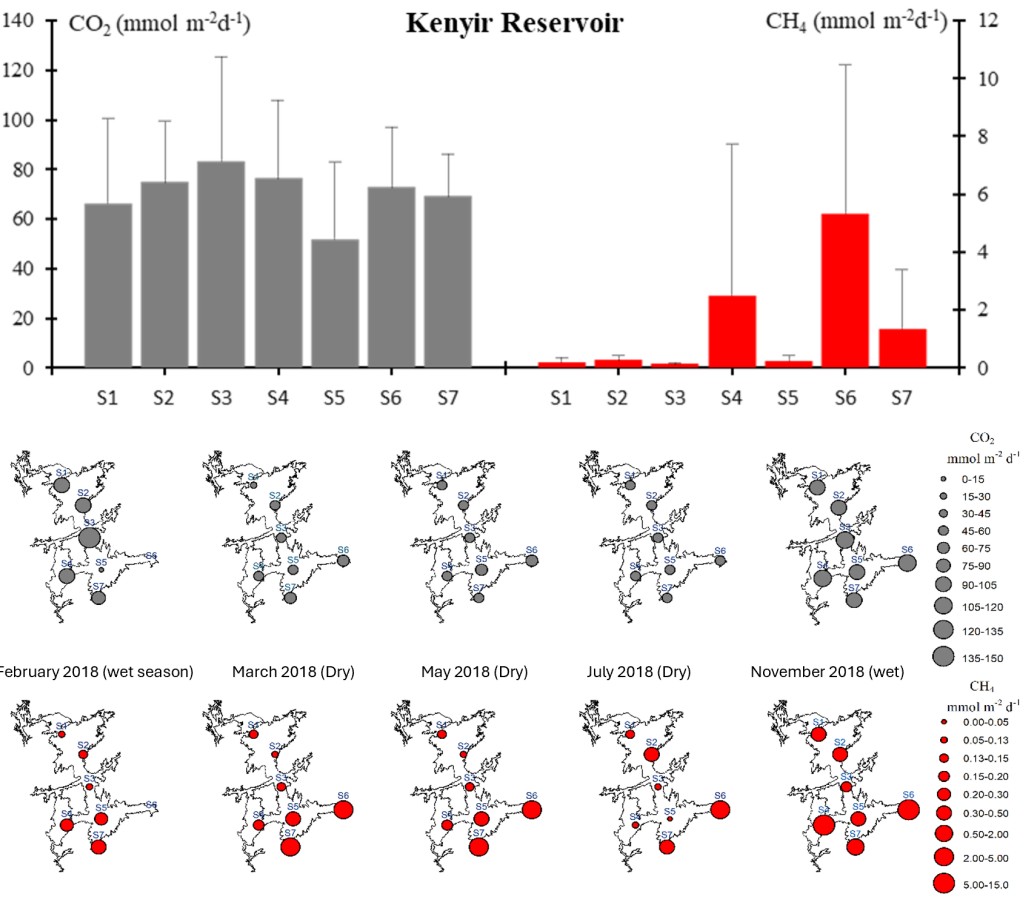

**Figure 2 The $CO_2$ and $CH_4$ emission fluxes pattern in Kenyir Reservoir.** Top panel: Averaged $CO_2$ and $CH_4$ emission fluxes at each sampling site. Middle and bottom panels: Temporal and spatial variations of $CO_2$ and $CH_4$ emission fluxes across Kenyir Reservoir, respectively.

during low flow. $CO_2$ concentrations ranged from 7 to 307 µM during high flow and 27 to 317 µM during low flow.

GHG flux measurements in estuary during high flow showed that $CH_4$ fluxes averaged 13 ± 5.0 mmol m$^2$ d$^{-1}$ (range: 4.3 to 21 mmol m$^2$ d$^{-1}$), while $CO_2$ fluxes averaged 20 ± 30 mmol m$^2$ d$^{-1}$, (range: −11 to 112 mmol m$^2$ d$^{-1}$, as shown in Fig. 4). In contrast, $CH_4$ fluxes remained relatively stable during low flow, averaging 4.1 ± 2.8 mmol m$^2$ d$^{-1}$ (range: 1.6 to 10 mmol m$^2$ d$^{-1}$). However, $CO_2$ fluxes were significantly higher (Kruskal–Wallis test $P < 0.05$), with an average of 232 ± 95 mmol m$^2$ d$^{-1}$ (range: 70 to 350 mmol m$^2$ d$^{-1}$).

Time-series measurements during neap and spring tides revealed notable differences in GHG concentrations and fluxes. During neap tide, $CH_4$ concentrations averaged 1.5 ± 0.6 µM, (range: 0.4 to 2.6 µM), while $CO_2$ concentrations averaged 56 ± 30 µM (range: from 20 to 109 µM). In contrast, $CH_4$ concentrations were lower during spring tide, averaging 0.6 ± 0.4 µM (range: below detection limit to 1.4 µM), and $CO_2$ concentrations were slightly higher, averaging 61 ± 38 µM (range: 24 to 126 µM).

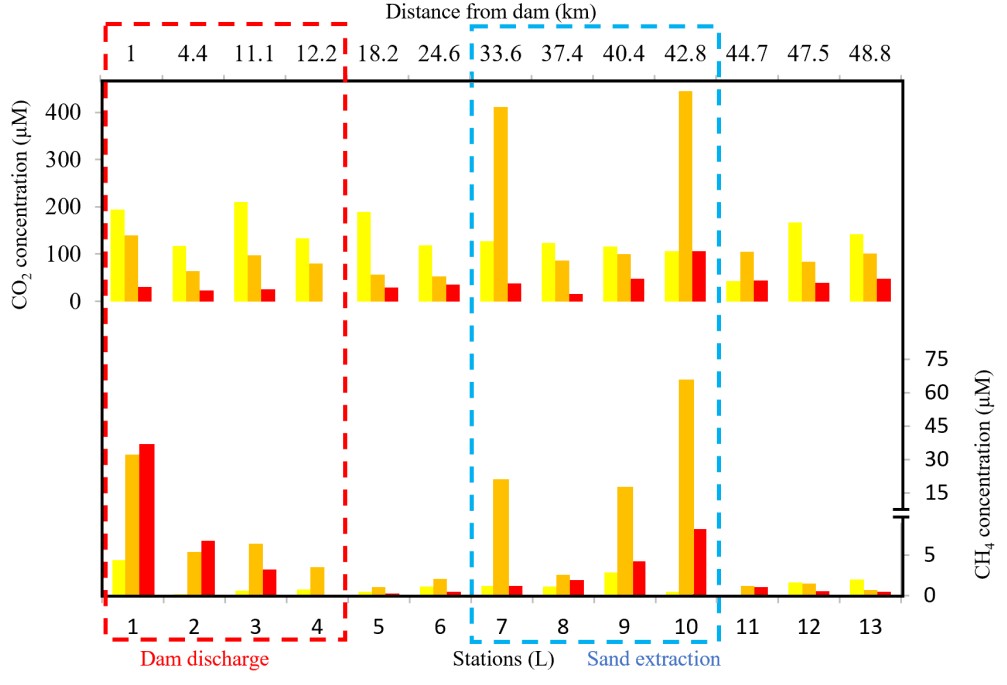

**Figure 3** **Longitudinal distribution of $CO_2$ and $CH_4$ concentration in Terengganu River.** Yellow, March 2018; orange, April 2018; red, April 2019.

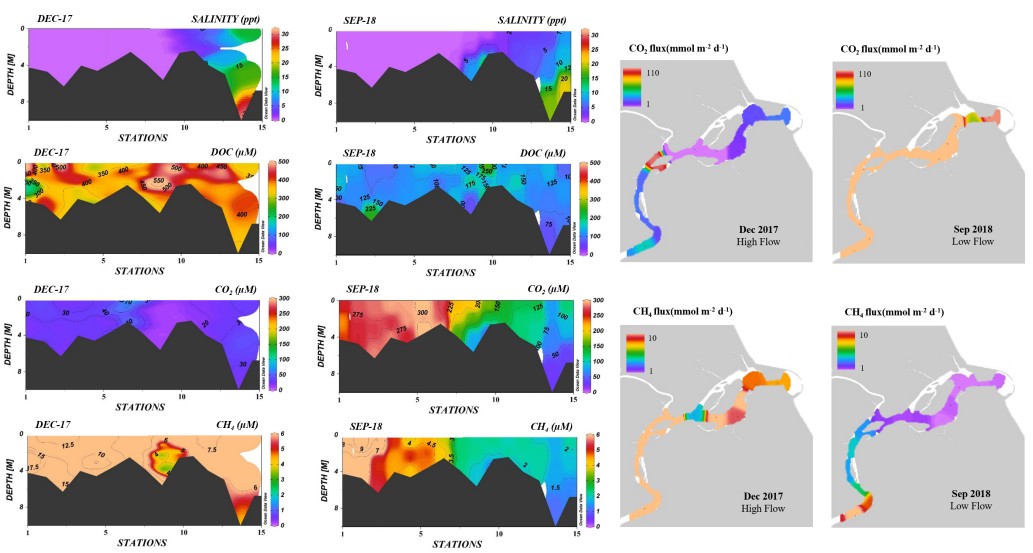

**Figure 4** **Longitudinal profiles of $CO_2$ and $CH_4$ concentrations and fluxes along the Terengganu River Estuary during high (Dec 2017) and low flow events (Sept 2018).**

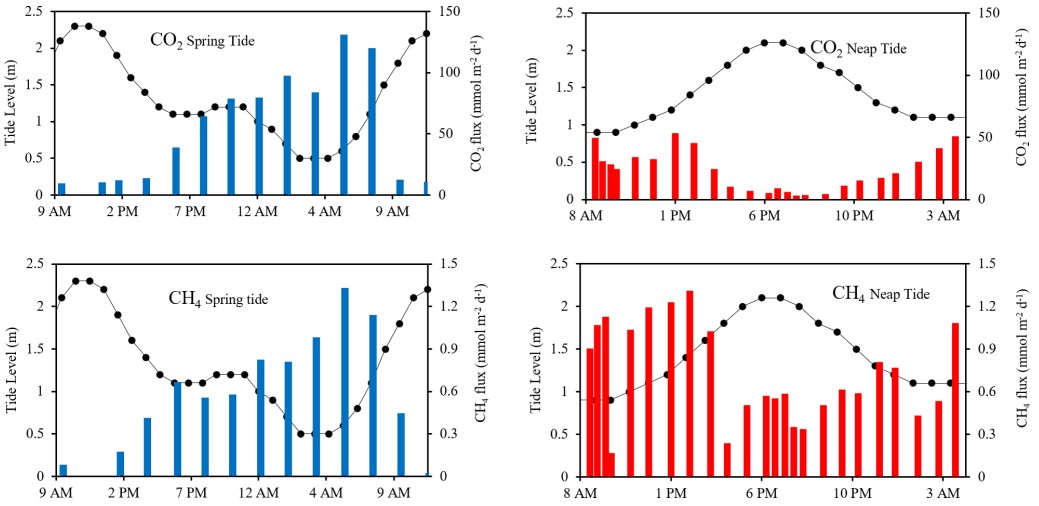

**Figure 5** **Diurnal variation of CO₂ and CH₄ fluxes in Terengganu River Estuary.** Black coloured line represents tide height in m; coloured bars represent the flux in mmol m⁻² d⁻¹.

Over a 24-hour sampling period, GHG fluxes during neap tide showed that $CH_4$ averaged $0.7 \pm 0.3$ mmol m² d⁻¹ (range: 0.2 to 1.3 mmol m² d⁻¹, Fig. 5), while $CO_2$ fluxes averaged $23 \pm 16$ mmol m² d⁻¹ (range: 3.2 to 53 mmol m² d⁻¹) During spring tide, the $CH_4$ fluxes averaged $0.6 \pm 0.4$ mmol m² d⁻¹ (range: −0.09 to 1.3 mmol m² d⁻¹), whereas $CO_2$ fluxes increased, averaging $55 \pm 44$ mmol m² d⁻¹ (range: 10 to 131 mmol m² d⁻¹).

## DISCUSSION

### Greenhouse emissions from Kenyir Reservoir

Temperature and dissolved oxygen profiles indicate that the Kenyir Reservoir is a meromictic system with stable thermal stratification year-round (Fig. S2). This thermal stratification drives oxygen stratification, isolating bottom waters from atmospheric exchange and resulting in anoxic conditions that enhance methane production through methanogenesis (*Davidson, Audet & Jeppesen, 2018*). Under these low-oxygen conditions, organic matter decomposition also contributes to elevated carbon dioxide levels. As a result, oxygen stratification enhances both $CH_4$ and $CO_2$ production in deeper waters (Fig. S3).

Statistically, the average $CO_2$ emission rate across all sites was significantly higher during the wet season (Kruskal–Wallis test $p < 0.05$). This increase may be attributed to stronger winds. In this study, monthly average wind speed in the Kenyir Reservoir was 1.3 times higher during the wet season than in the dry season (Table S4). This finding aligns with the findings of *Sawakuchi et al. (2017)* that windier conditions promote surface water mixing and enhance the diffusion of dissolved GHG into the atmosphere.

Ebullition was observed in the shallow southern region near S7 (Fig. S2), likely due to shallow water. Previous studies (*Bastviken et al., 2004*; *Gerardo-Nieto et al., 2017*; *West, Creamer & Jones, 2016*; *DelSontro et al., 2010*) have shown that ebullition is more common

in littoral zones. In the Kenyir Reservoir, active ebullition occurred near to S7, where water depths were less than 40 m.

The overall water level increase in Kenyir Reservoir during the wet season was associated with a lower ebullition rate, as higher hydrostatic pressure suppresses gas bubble formation, similar to the observation of *Ostrovsky et al. (2008)* in Lake Kinneret. We assume that GHG production from the bottom sediments remains consistent year-round due to the persistent anoxic conditions at the reservoir bottom (Fig. S4). However, intense wind forcing during the wet season (Table S4) enhances internal lake motion, disrupting the ascent of GHG bubbles. *McGinnis et al. (2006)* demonstrated that increased horizontal currents break larger bubbles into smaller ones, increasing the total surface area available for bubble redissolution before they reach the surface. Consequently, these processes result in fewer GHG bubbles captured by submersible trapping funnels in S7 during the wet season.

Table 1 presents the $CO_2$ and $CH_4$ emissions from 39 freshwater reservoirs and lakes latitudinally from north to south. Kenyir Reservoir ranks 9th and 8th for $CH_4$ and $CO_2$ among the global freshwater GHG emissions references in Table 1. The complied data showed that tropical reservoirs release more GHG than boreal, temperate, and subtropical reservoirs. For instance, Tucurui Reservoir in Brazil measured 0–180 mmol $CH_4$ $m^{-2}$ $d^{-1}$ and 30–3,242 mmol $CO_2$ $m^{-2}$ $d^{-1}$ in 1993 (*Dos Santos et al., 2006*). Curuá-Una reservoir measured $CH_4$ emissions of 0.1–7.0 mmol $m^{-2}$ $d^{-1}$ and $CO_2$ emissions of 0.6–1,612 mmol $m^{-2}$ $d^{-1}$ in 2017 (*Paranaíba et al., 2021*), which were nearly ten times higher than Lake Allatoona, USA, in the sub-tropical reservoir that recorded 11.7 mmol $CH_4$ $m^{-2}$ $d^{-1}$ and 54.8 mmol $CO_2$ $m^{-2}$ $d^{-1}$ in 2012 (*Bevelhimer et al., 2016*).

Tropical reservoirs exhibit higher GHG emissions (0–180 mmol $CH_4$ $m^{-2}$ $d^{-1}$ and −36–3,243 mmol $CO_2$ $m^{-2}$ $d^{-1}$) than reservoirs in other climates (0–18.3 mmol $CH_4$ $m^{-2}$ $d^{-1}$ and −5.5–213 mmol $CO_2$ $m^{-2}$ $d^{-1}$). This difference is largely due to greater organic carbon in tropical reservoirs and higher surface runoff, which continuous supplies organic matter, fueling bacterial metabolism and enhancing GHG production (*Tranvik et al., 2009*). Table 1 also shows higher GHG emissions are commonly observed in young reservoirs (less than 10 years). For example, in the first year after damming, annual diffusive $CO_2$ and $CH_4$ emissions in reservoirs, such as Nam Theun 2, Nam Leuk (Laos), Petit-Saut (French Guiana), and Tucurui (Brazil), were approximately 1 to 10 times greater than in later years. Despite being more than 30 years old, the Kenyir Reservoir continues to exhibit high $CO_2$ and $CH_4$ emission than other reservoirs with similar age, such as the Batang Ai Reservoir (*Soued & Prairie, 2020*) and Reservoir Lokka (*Huttunen et al., 2004*).

### Greenhouse gas emissions downstream of a dammed river

High GHG concentrations in the Terengganu River were observed near the dam discharge outlets (4.4–36.5 µM $CH_4$ and 30.3–194 µM $CO_2$), extending up to 12 km downstream (0.7–3.6 µM $CH_4$ and 80–133 µM $CO_2$) from L1 to L4 (Fig. 3). In these upstream sections, $CH_4$ concentrations were negatively correlated with dissolved oxygen ($r = -0.6$, $p < 0.05$), suggesting that the elevated $CH_4$ levels originated from the hypoxic bottom waters of the reservoir. Consistently high $CH_4$ concentrations were recorded near the reservoir's bottom (mean $621 \pm 187$ µM, $n = 20$), where oxygen depletion provides favorable conditions

**Table 1  Comparison of Kenyir Reservoir CH$_4$ and CO$_2$ emissions with global Reservoirs.**

| References | Region | Reservoir | Country | Latitude | Age during measurement year | Surface area km$^2$ | Diffusive flux CH$_4$ mmol m$^{-2}$ d$^{-1}$ | Diffusive flux CO$_2$ mmol m$^{-2}$ d$^{-1}$ | C emission Gg C yr$^{-1}$ |
|---|---|---|---|---|---|---|---|---|---|
| *Huttunen et al. (2004)* | Temperate | Reservoir Lokka | Finland | 68°N | 28 | 216-417 | 0.33–7.4 | 11–73 | 123 |
| *Huttunen et al. (2004)* | Temperate | Reservoir Porttipahta | Finland | 68°N | 25 | 34–214 | 0.16–0.3 | 20–52 | 83.7 |
| *Roehm & Tremblay (2006)* | Temperate | La Grande-2 | Canada | 54°N | 26 | 2815 | 0–0.16 | 15 (1–148) | 845.6 |
| *Roehm & Tremblay (2006)* | Temperate | La Grande-3 | Canada | 54°N | 21 | 2,536 | 9–34 | | 325.9 |
| *Martinez-Cruz et al. (2020)* | Temperate | Lake Dagow | Germany | 53°N | | 0.24 | 2.3 | 6 | 0.02 |
| *Martinez-Cruz et al. (2020)* | Temperate | Lake Stechlin | Germany | 53°N | 55 | 4.25 | 0.09 | −5.5 | −0.4 |
| *Brothers, Prairie & delGiorgio (2012)* | Temperate | Eastmain-1 | Canada | 52°N | 2 | 603 | 1.4 | 13.6 | |
| *Descloux et al. (2017)* | Temperate | Eguzon Reservoir | France | 47°N | 85 | 2.7 | 0.03–5.6 | | 0.02 |
| *Demarty et al. (2009)* | Temperate | Rivière-des-Prairies | Canada | 46°N | 79 | 42.3 | 0–0.4 | 15 (-5–213) | 10.2 |
| *Samiotis et al. (2018)* | Temperate | Polyfytos Reservoir | Greece | 40°N | 4 | 21.9 | 0–6.7 | 4.1–45.1 | 6.9 |
| *Samiotis et al. (2018)* | Temperate | Ilarion Dam | Greece | 40°N | 42 | 74 | 0–11.6 | 1.1–41 | 25.6 |
| *Beaulieu et al. (2020)* | Subtropical | Acton Lake | USA | 40°N | 59 | 2.4 | 9.3–18.3 | | 0.2 |
| *McClure et al. (2018)* | Subtropical | Falling Creek Reservoir | USA | 38°N | 117 | 0.119 | 0.7 | −4.6–135 | 0.1 |
| *Bevelhimer et al. (2016)* | Subtropical | Watts Bar Lake | USA | 36°N | 70 | 176 | 0.5 | 62.7 | 177.8 |
| *Bevelhimer et al. (2016)* | Subtropical | Guntersville Lake | USA | 36°N | 73 | 279 | 1.3 | 40.8 | 185.0 |
| *Bevelhimer et al. (2016)* | Subtropical | Fontana Lake | USA | 36°N | 68 | 43 | 0.4 | 22.6 | 15.7 |
| *Bevelhimer et al. (2016)* | Subtropical | Hartwell Lake | USA | 35°N | 50 | 226 | 1.4 | 26.5 | 98.1 |
| *Bevelhimer et al. (2016)* | Subtropical | Lake Allatoona | USA | 34°N | 63 | 49 | 11.7 | 54.8 | 46.5 |
| *Liu et al. (2016)* | Subtropical | Ross Barnett Reservoir | USA | 33°N | 45 | 134 | | 26.8 | 57.6 |
| *Chen et al. (2011)* | Subtropical | Three Gorges Reservoir | China | 31°N | 5 | 1,080 | 0.34 ± 0.3 | 76 ± 10.8 (76–213) | 1327.6 |
| *Xing et al. (2005)* | Subtropical | Lake Donghu | China | 31°N | | 19.03 | 1.5 ± 1.2 | 7.6 ± 3.6 | 2.5 |
| *Chanudet et al. (2011)* | Tropical | Nam Ngum | Laos | 19°N | 39 | 350 | 0.1–0.6 | −21.2 to −2.7 | 112.0 |
| *Deshmukh (2013)* | Tropical | Nam Theun 2 Reservoir | Laos | 18°N | 3 | 450 | 0–157 | 68 ± 51 | 500.8 |
| *Chanudet et al. (2011)* | Tropical | Nam Leuk | Laos | 18°N | 11 | 13 | 0.8–12 | −11–38 | 6.6 |
| *Prasad et al. (2013)* | Tropical | Dowleiswaram dam | India | 17°N | 24 | 70 | | 10–473 | 47.2 |
| *Guérin et al. (2006)* | Tropical | Petit-Saut Reservoir | France | 5°N | 9 | 365 | 4.6 ± 4.9 (0.1–7.7) | 119 ± 98 (102–133) | 708 |
| *Soued & Prairie (2020)* | Tropical | Batang Ai Reservoir | Malaysia | 1°N | 33 | 68.4 | 0.03–3.7 | −31–80 | 8.7 |
| *Guérin et al. (2006)* | Tropical | Balbina | Brazil | 2°S | 16 | 2,360 | 2.1 ± 3 (0.31–21) | 76 ± 46 | 2913 |
| *Paranaíba et al. (2021)* | Tropical | Curuá-Una | Brazil | 3°S | 40 | 72 | 0.08–7 | 3.4–1612 | 107 |
| *Dos Santos et al. (2006)* | Tropical | Tucurui | Brazil | 4°S | 9 | 2,430 | 0.002–180 | 10.4–3243 | 7606 |
| *Almeida et al. (2016)* | Tropical | Ecological Station of Serido | Brazil | 7°S | 70 | 0.2 | 11.8 ± 4.1 | 0.04 | |
| *Macklin et al. (2018)* | Tropical | Palasari Reservoir | Indonesia | 8°S | 27 | 1 | 372 | 3.5 | |
| *Guérin et al. (2006)* | Tropical | Samuel | Brazil | 9°S | 17 | 559 | 5.0 ± 5.9 | 976 ± 1213 | 8,780 |
| *Teodoru et al. (2015)* | Tropical | Cahora Bassa | Africa | 16°S | 39 | 2,675 | 0.08 | −8.1 | −347 |
| *Teodoru et al. (2015)* | Tropical | Itezhi Tezhi | Africa | 16°S | 35 | 365 | 1.6 | 16.7 | 101 |
| *Paranaíba et al. (2021)* | Tropical | Furnas Dam | Brazil | 21°S | 54 | 1,342 | 0.001–21.5 | −36–90 | 117 |
| *Paranaíba et al. (2021)* | Tropical | Funil Reservoir | Brazil | 21°S | 13 | 40 | 0.0002–2.1 | −0.05–1.7 | 0.1 |
| *Paranaíba et al. (2021)* | Tropical | Chapéu D'Úvas | Brazil | 22°S | 23 | 12 | 0.02–19 | −26–32 | 1.3 |
| This study | Tropical | Kenyir Reservoir | Malaysia | 5°N | 31 | 369 | 0.02–12.5 | 5.6–143.8 | 451.9 |
for methanogenesis (*Conrad, 2020*; *Müller et al., 2019*). From site L1 to L4, immediately downstream of the reservoir outlets, $CH_4$ concentration decreased by approximately 80% (Fig. 3). This decline was likely driven by a combination of dilution, atmospheric evasion, and microbial oxidation (*DelSontro et al., 2016b*; *Guérin & Abril, 2007*; *Kemenes, Forsberg & Melack, 2007*). Further downstream at L5 and L6, $CH_4$ concentrations dropped below five μM.

The substantial increase in $CO_2$ and $CH_4$ concentration in the middle segment of the Terengganu River (L7 to L10, Fig. 3) is most likely linked to in-stream sand extraction activities. At least four active in-stream sand extraction operations were observed during the sampling campaign along this river section. These activities involved submersible pumps extracting riverbed materials comprising gravel, sand, silt, and mud, resulting in the mineralization of resuspended particulate organic matter, releasing substantial $CO_2$ and $CH_4$ into the overlying water column. In addition, the disturbance of riverbed sediments may have promoted the degassing of $CH_4$ and $CO_2$ from porewater (*Marcon et al., 2022*). *Qin et al. (2020)* also reported that in-stream sand extraction disrupts the carbon sequestration potential of riparian areas, significantly reducing $CO_2$-fixing microbial communities and leading to increased GHG emissions. Beyond its biogeochemical impacts, sand extraction disturbs sediment supply and transport equilibrium, triggering morphological alteration that causes irreversible changes in the watershed characteristics (*Padmalal & Maya, 2014*). The resuspension of sediments increases turbidity in the water column (*Ashraf et al., 2011*). Over time, reduced light penetration limits photosynthesis and oxygen production, promoting the anaerobic condition in the benthic layer, which enhances microbial decomposition of organic matter, further increasing $CO_2$ and $CH_4$ production.

The Terengganu River is a source of $CO_2$ and $CH_4$ to the atmosphere with fluxes ranging from 0.7 to 136 mmol $CO_2$ $m^{-2}$ $d^{-1}$ and 0.06 to 20 mmol $CH_4$ $m^{-2}$ $d^{-1}$, respectively. In-stream sand extraction is an important contributor to the river's $CH_4$, and $CO_2$ flux. Overall, sand extraction activity alone has contributed more than 50% of Terengganu River's total carbon ($CO_2$+$CH_4$) emission (Table 2). Downstream discharge water from the dam is the second largest $CH_4$ source, contributing 30% of the diffusive flux to the atmosphere.

The median concentration of $CO_2$ (99 μM) and $CH_4$ (1.6 μM) in the Terengganu River was at least 6 and 8,000 times higher than atmospheric levels ($\sim$17 μM $CO_2$; $\sim$0.002 μM $CH_4$), respectively. Overall, $CO_2$ and $CH_4$ levels in the Terengganu River fall within the typical range of global tropical rivers (42–337 μM $CO_2$; 0.4–350 μM $CH_4$) but are at least an order of magnitude lower than those in dammed rivers and peat-draining rivers (Table 3). In terms of diffusive flux, Terengganu River $CO_2$ (0.7 to 136 mmol $m^{-2}$ $d^{-1}$) and $CH_4$ (0.06 to 20 mmol $m^{-2}$ $d^{-1}$) fluxes are comparable to those reported for Wohlen Reservoir (*DelSontro et al., 2016b*), River Tay (*Harley, 2013*), Tianjin River (*Hu et al., 2018*), and as well as Kariba Dam (*Teodoru et al., 2015*). However, they are one to two orders of magnitude lower than fluxes reported for tropical reservoirs and rivers in Brazil (*Abril et al., 2005*; *Guérin et al., 2006*; *Kemenes, Forsberg & Melack, 2007*; *Kemenes, Forsberg & Melack, 2011*), Indonesia (*Wit et al., 2015*), and Malaysia (*Müller et al., 2015*).

**Table 2** Total diffusive $CO_2$ and $CH_4$ flux in Terengganu River and the estimated contributions from specific anthropogenic activities.

| Sampling period | $CO_2$ (mmol m$^{-2}$ d$^{-1}$) median ± std | Total river emission (Mmol d$^{-1}$) | Contribution (%) | | |
|---|---|---|---|---|---|
| | | | Dam discharge | Sand extraction | Other |
| Mar-18 | 7.5 ± 7.1 | 0.08 | 8 | 49 | 43 |
| Apr-18 | 26 ± 42 | 0.43 | 10 | 71 | 18 |
| Apr-19 | 37 ± 14 | 0.40 | 24 | 32 | 45 |
| | | Average contribution (%) | 14 | 51 | 35 |
| | $CH_4$ (mmol m$^{-2}$ d$^{-1}$) | | | | |
| Mar-18 | 0.5 ± 3.0 | 0.01 | 55 | 38 | 7 |
| Apr-18 | 1.0 ± 5.7 | 0.03 | 16 | 79 | 5 |
| Apr-19 | 0.3 ± 0.4 | 0.003 | 18 | 47 | 35 |
| | | Average contribution (%) | 30 | 55 | 15 |

## Greenhouse gas emission dynamics in Terengganu River Estuary

The average $CH_4$ concentrations in the estuary were significantly higher during high flow (9.2 ± 4.4 µM) than during low flow (3.6 ± 2.1 µM). Elevated $CH_4$ levels observed during the high flow period in December 2017 were likely driven by substantial rainfall-induced urban runoff. Increased dissolved organic carbon during these events may have contributed to higher $CH_4$ concentration in the estuary (Fig. 4). Previous studies have shown that river segments draining urban areas are important $CH_4$ sources (*Tang et al., 2021*). $CO_2$ and $CH_4$ removal were observed from sites E7 to E15 during low flow periods, probably due to mixing with seawater in the lower estuary (Fig. 4).

Seasonally, the largest range of $CO_2$ fluxes was observed during the dry season (September 2018), ranging from 70 to 350 mmol m$^{-2}$ d$^{-1}$ compared to the wet season (December 2017), which ranged from −11 to 110 mmol m$^{-2}$ d$^{-1}$ (Fig. 4). In contrast, average $CH_4$ fluxes were higher in December 2017 (13.1 mmol m$^{-2}$ d$^{-1}$) than in September 2018 (4.1 mmol m$^{-2}$ d$^{-1}$) ($p < 0.05$). Overall, $CO_2$ and $CH_4$ efflux in TRE displayed a decreasing trend from the upper to the lower reaches (Fig. 4). The elevated $CH_4$ emissions in the freshwater portion during December 2017 were likely due to inputs from nearby tributaries, stormwater discharge outlets, and non-point sources. Additionally, breakwater structure and land reclamation near the estuarine inlet may have altered water column mixing, potentially facilitating *in situ* GHG production (Fig. 4). This observation is consistent with *Looman, Maher & Santos (2021)*, who reported increased localized outgassing due to altered watercourse hydrodynamics. Conversely, the decline in $CO_2$ and $CH_4$ fluxes in the lower estuary was likely driven by degassing from tidal oscillation and dilution with seawater.

The time-series $CO_2$ flux in the estuary varied primarily with tidal height rather than diurnal fluctuations (Fig. 5). During the spring tide, the maximum average $CO_2$ flux
**Table 3  Greenhouse gases diffusive fluxes at the air-water interface of rivers around the world.**

| Location | River/Reservoir Immediate Downstream (year of measurement) | Dissolved surface GHGs ($\mu mol\ L^{-1}$) | | Diffusive flux ($mmol\ m^{-2}\ d^{-1}$) | | References |
|---|---|---|---|---|---|---|
| | | $CH_4$ | $CO_2$ | $CH_4$ | $CO_2$ | |
| Switzerland | Wohlen Reservoir (2013) | 0.1–1.3 | – | 0.13–3.0 | – | *DelSontro et al. (2016a)* |
| Scotland | River Tay (2010) | – | – | 0.1–1.0 | 12–58 | *Harley (2013)* |
| China | Tianjin River (2015) | $1.4 \pm 1.2$ | $38 \pm 8.6$ | $1.7 \pm 1.6$ | $20 \pm 10$ | *Hu et al. (2018)* |
| Brazil | Kariba Dam (2013) | <0.1 | 106 | – | – | *Teodoru et al. (2015)* |
| | Itezhi Tezhi Dam (2013) | <0.1 | 42 | 1.2 | 13 | |
| | Cabora Bassa Dam (2013) | – | – | – | 34 | |
| | Petit-Saut Reservoir (2003) | $2.5 \pm 2.6$ | $108 \pm 63$ | 59 | 1,003 | *Abril et al. (2005)* |
| | Petit-Saut Reservoir (2005) | 48 | 311 | $84 \pm 38$ | $802 \pm 364$ | *Guérin et al. (2006)* |
| | Samuel Reservoir (2004) | 40 | 337 | $12 \pm 13$ | $1,494 \pm 963$ | |
| | Balbina Reservoir (2004) | $77 \pm 7$ | $203 \pm 27$ | $114 \pm 66$ | $412 \pm 95$ | |
| | Balbina Reservoir (2005) | 0.4-140 | – | 105.4 | – | *Kemenes, Forsberg & Melack (2007)* |
| | Balbina Reservoir (2006) | – | 161 | – | 109 | *Kemenes, Forsberg & Melack (2011)* |
| | | | | (0.5–287) | | |
| Indonesia | Batanghari River (2009) | $98 \pm 0.7$ | – | $20 \pm 18$ | – | *Wit et al. (2015)* |
| | Indragiri River (2013) | $236 \pm 22$ | – | $232 \pm 61$ | – | |
| | Siak River (2013) | $350 \pm 22$ | – | $320 \pm 61$ | – | |
| Malaysia | Maludam River (2014) | $319 \pm 37$ | – | $289 \pm 155$ | – | *Müller et al. (2015)* |
| | Maludam River (2015) | $343 \pm 6$ | – | $125 \pm 136$ | – | |
| | Terengganu River (2018) | 9.1 | 92 | 0.8 | 14 | This study, median |
| | | 0.3–65 | 15–445 | 0.06–20 | 0.7–136 | range |

reached 40 mmol $m^{-2}$ $d^{-1}$ during the daytime and 74 mmol $m^{-2}$ $d^{-1}$ at night. In contrast, during the neap tide, $CO_2$ flux ranged from 3 to 53 mmol $m^{-2}$ $d^{-1}$, peaking at 51 mmol $m^{-2}$ $d^{-1}$ at 0400 h. The average $CO_2$ flux during the daytime was 31 mmol $m^{-2}$ $d^{-1}$, while at night it was 17 mmol $m^{-2}$ $d^{-1}$.

Similarly, $CH_4$ fluxes were strongly influenced by tidal conditions. During spring tides, the average $CH_4$ flux reached 0.7 mmol $m^{-2}$ $d^{-1}$ during the ebb tide, significantly higher than the 0.008 mmol $m^{-2}$ $d^{-1}$ recorded during the flood tide. For neap tides, $CH_4$ emissions showed a comparable amplitude to those during spring tides, ranging from 0.2 to 1.3 mmol $m^{-2}$ $d^{-1}$, with a peak of 1.3 mmol $m^{-2}$ $d^{-1}$ at 1400 h. The average daytime emission was 0.9 mmol $m^{-2}$ $d^{-1}$, while nighttime average was 0.6 mmol $m^{-2}$ $d^{-1}$.

Although these differences suggest a possible diel component, tidal height appears to be the dominant factor controlling emissions. Higher tidal amplitudes during spring tides correspond to longer ebb tide duration (*Mao et al., 2004*), which enhances tidal damping and facilitates more efficient material exchange and seaward transport (*Arueira et al., 2022*; *Geyer, Chant & Houghton, 2008*; *Siegle et al., 2009*). During the ebb tide, as water levels

in the estuary drop, the reduced water column allows trapped $CO_2$ in the organic-rich sediment to be released more effectively.

Similarly, $CH_4$ emission in estuaries strongly depends on tidal height variation, occurring predominantly at low tide and primarily in shallow water depth systems (*Ferrón et al., 2007*; *Grunwald et al., 2007*; *Harley, 2013*). *Harley (2013)* showed that a falling tide can partially replace the estuary water column with $CH_4$-supersaturated freshwater from the upper estuary. *Ferrón et al., (2007)* and *Grunwald et al. (2007)* also observed that $CH_4$ concentration increases as tidal amplitude and salinity decrease, attributing these changes to allochothonous inputs, urban effluent discharge, *in situ* production, or the drainage of $CH_4$-rich pore water from tidal flats.

Table 4 compares $CH_4$ diffusive flux in the studied estuary with other river estuaries worldwide. The $CH_4$ flux in the estuary is comparable to the Jiulong River Estuary, China (*Li et al., 2021*), Adyar Estuary, India (*Nirmal Rajkumar et al., 2008*) and Scheldt Estuary, United Kingdom (*Middelburg et al., 1996*), spanning sub-tropical, tropical, and temperate regions. However, the $CO_2$ diffusive flux in Terengganu River estuary exceeds the range reported for tropical river estuaries. It is an order of magnitude higher than the sub-tropical estuaries in China (*Yu et al., 2013*) and the USA (*Jiang, Cai & Wang, 2008*). High $CH_4$-emitting estuaries in Table 4 share two key similarities: elevated suspended particulate matter levels and an external forcing mechanism that enhances $CH_4$ production. These external factors include tidal induced resuspension (*Li et al., 2021*), prolonged water residence times due to reduced river discharge (*Nirmal Rajkumar et al., 2008*), freshwater inflow influced by sewage (*Middelburg et al., 1996*), and hydrodynamic alterations from breakwater and land reclamation (this study). Additonal contributing factors include pollution, monsoonal influences, and estuarine morphological changes (*Borges, Abril & Bouillon, 2018*; *Gupta et al., 2009*).

The total GHG emissions from the Terengganu River catchment are estimated at 572 Gg $CO_2$-equivalent per year, with 85% $CO_2$ and 15% $CH_4$. This estimate applies to a global warming potential of 27 for $CH_4$, based on a value from the IPCC Sixth Assessment Report (AR6). The catchment spans 386.6 $km^2$, with the upper freshwater section (Kenyir Reservoir) covering 369 $km^2$ and containing highly supersaturated dissolved GHG that influences the 61.5 km-long Terengganu River and its estuary. The river occupies ∼10 $km^2$, while the estuary spans 7.6 $km^2$. Average GHG flux from each section was multiplied by their respective surface areas, revealing that 94% of total emissions originated from the reservoir, 5.5% from the estuary, and only 0.5% from the river. Despite its smaller surface area than the river, the estuary contributes a disproportionately high share of emissions due to its higher organic matter content and reduced water turnover, exacerbated by the semi-enclosed breakwater that further elevated $CH_4$ emissions during the wet season.

## CONCLUSIONS

Human activities significantly shape the spatial distribution of $CH_4$ and $CO_2$ concentrations within the Terengganu River system (Fig. 6). High $CH_4$ concentrations were observed downstream of the dam (S6), in the shallow littoral zone of the Kenyir Reservoir (S7), near

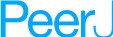

**Table 4  CH$_4$ and CO$_2$ concentration and flux in estuaries around the world.**  Data represent the range (mean).

| Study area | Country | Latitude | Dominant sources and processes | Measurement year | CH$_4$ Concentration range ($\mu$M) | CO$_2$ Concentration range ($\mu$M) | CH$_4$ Flux range (mmol m$^{-2}$ d$^{-1}$) | CO$_2$ Flux range (mmol m$^{-2}$ d$^{-1}$) | References |
|---|---|---|---|---|---|---|---|---|---|
| Pulicat Lake Estuary | India | 14°N | Lateral inputs | na | 0.09–0.50 (0.24) | | 0.05–0.23 | | *Shalini et al. (2006)* |
| New River Estuary | USA | 34°N | Lateral inputs | 2016 | | | | −6.6–98.4 | *Van Dam, Edson & Tobias (2019)* |
| Neuse River Estuary | USA | 35°N | Lateral inputs | 2016 | | | | −2.7–98.4 | *Van Dam, Edson & Tobias (2019)* |
| Hugli Estuary | India | 22°N | Lateral inputs | 2014 | | | | (88.8) | *Akhand et al. (2016)* |
| Hooghly Estuary | India | 21°N | Lateral inputs | 1999 | | | | −2.8–84.4 | *Mukhopadhyay et al. (2002)* |
| Betsibika Estuary (D) | Madagascar | 18°S | Lateral inputs | 2005 | | 11–62.5 | | (9.1) | *Ralison et al. (2008)* |
| Guadalquivir Estuary | Spain | 37°N | Tidal flats | 2017 | (0.05) | (39.2) | (0.12) | (60) | *Sierra et al. (2020)* |
| Guadiana Estuary | Spain | 37°N | Tidal flats | 2017 | (0.05) | (24.3) | (0.11) | (40) | *Sierra et al. (2020)* |
| Tinto-Odiel Estuary | Spain | 37°N | Tidal flats | 2017 | (0.04) | (23.7) | (0.07) | (11.7) | *Sierra et al. (2020)* |
| Matla Estuary | India | 22°N | Tidal flats | 2014 | | | | (6.3) | *Akhand et al. (2016)* |
| Rajang River Estuary (D) | Malaysia | 2°N | Tidal flats | 2016 | | (114) | | 15.9–68.2 (45.4) | *Müller et al. (2019)* |
| Rajang River Estuary (W) | Malaysia | 2°N | Tidal flats | 2016 | | (122) | | 15.9–54.5 (40.9) | *Müller et al. (2019)* |
| Lupar River Estuary (D) | Malaysia | 1°N | Tidal flats | 2013 | 0.01–0.04 (0.02) | | <0 | | *Müller et al. (2016)* |
| Lupar River Estuary (W) | Malaysia | 1°N | Tidal flats | 2013 | 0–0.06 (0.02) | | <0 | | *Müller et al. (2016)* |
| Saribas River Estuary (D) | Malaysia | 1°N | Tidal flats | 2013 | 0.01–0.06 (0.03) | | <0 | | *Müller et al. (2016)* |
| Saribas River Estuary (W) | Malaysia | 1°N | Tidal flats | 2013 | 0.01–0.07 (0.02) | | <0 | | *Müller et al. (2016)* |
| Lupar River Estuary (D) | Malaysia | 1°N | Tidal flats | 2014 | | (101) | | (312) | *Müller et al. (2016)* |
| Lupar River Estuary (W) | Malaysia | 1°N | Tidal flats | 2014 | | (75.6) | | | *Müller et al. (2016)* |
| Saribas River Estuary (D) | Malaysia | 1°N | Tidal flats | 2014 | | (92.0) | | (200) | *Müller et al. (2016)* |
| Saribas River Estuary (W) | Malaysia | 1°N | Tidal flats | 2014 | | (91.4) | | | *Müller et al. (2016)* |
| Furo do Meio in Caeté Estuary | Brazil | 1°S | Tidal flats | 2017 | | | (0.86) | (174) | *Call et al. (2018)* |
| Altamaha Sound | USA | 32°N | Tidal flats | 2004 | | | | (69.3) | *Jiang, Cai & Wang (2008)* |
| Tagus Estuary | Portugal | 38°N | Tidal dynamics | 2007 | | | | 24.5–65.5 | *Oliveira et al. (2018)* |
| Changjiang River Estuary | China | 29°N | Tidal dynamics | 2006 | 0–0.09 (0.10) | | (0.06) | | *Zhang et al. (2008)* |
| Pearl River Estuary | China | 26°N | Tidal dynamics | 2003 | 0.02–2.98 (2.9) | 28.2–110 | | | *Chen et al. (2008)* |
| Jiulong River Estuary | China | 25°N | Tidal dynamics | 2018 | 0.06–1.7 | | 0.19–18.6 | | *Li et al. (2021)* |
| Columbia River Estuary | USA | 46°N | Landuse activity | 2013 | 0.27–0.73 (0.45) | | | | *Pfeiffer-Herbert et al. (2019)* |
| Oregon Estuary | USA | 44°N | Landuse activity | 1982 | | | 0–1.30 | | *De Angelis Marie & Lilley Marvin (1987)* |
| Mekong Delta | Vietnam | 17°N | Landuse activity | 2004 | 0–2.22 | 9.5–167 | 0.04–0.2 | 105–135 | *Borges, Abril & Bouillon (2018)* |
| Scheldt Estuary | UK | 51°N | Sewage | 1991 | | | 0.14–189 (21.6) | 17.5–227 (120) | *Middelburg et al. (1996)* |
| Hudson River Estuary | USA | 42°N | Sewage | 1991 | 0.05–0.94 (0.9) | | (0.30) | | *DeAngelis & Scranton (1993)* |
| Changjiang River Estuary | China | 29°N | Sewage | 2010 | | 7.2–86 | | 0–230 (61) | *Yu et al. (2013)* |
| Adyar Estuary (W) | India | 13°N | Sewage | 2001 | | | (23.1) | | *Purvaja & Ramesh (2001)* |
| Adyar Estuary (W) | India | 13°N | Sewage | 2004 | 0.03–5.62 | | 0.04–11.0 | | *Nirmal Rajkumar et al. (2008)* |
| Cochin Estuary (D) | India | 10°N | Sewage | 2005 | | 51.6–105 | | 65–267 | *Gupta et al. (2009)* |
| Brisbane River Estuary | Australia | 27°S | Sewage | 2012 | 0.08–0.31 | | 0.25–2.2 | | *Sturm et al. (2017)* |
| Brisbane River Estuary | Australia | 27°S | Sewage | 2012 | 0.03–0.58 (0.5) | | 0.02–1.7 | | *Musenze et al. (2013)* |
| Godavari River Estuary (D) | India | 19°N | Urbanization | 2009 | | 9–1339 | | 80–150 | *Sarma et al. (2011)* |
| Terengganu River Estuary (D) | Malaysia | 5°N | Urbanization | 2018 | 1.60–9.6 (4.0) | 75–312 (212) | 1.6–10.1 | 70–350 (233) | This study |
| Terengganu River Estuary (W) | Malaysia | 5°N | Urbanization | 2018 | 1.90–14.7 (5.7) | 9–271 (135) | 2.5–21.1 | −11–391 (179) | This study |
| Caboolture River Estuary | Australia | 27°S | Urbanization | 2012 | | | | (78) | *Jeffrey et al. (2018)* |

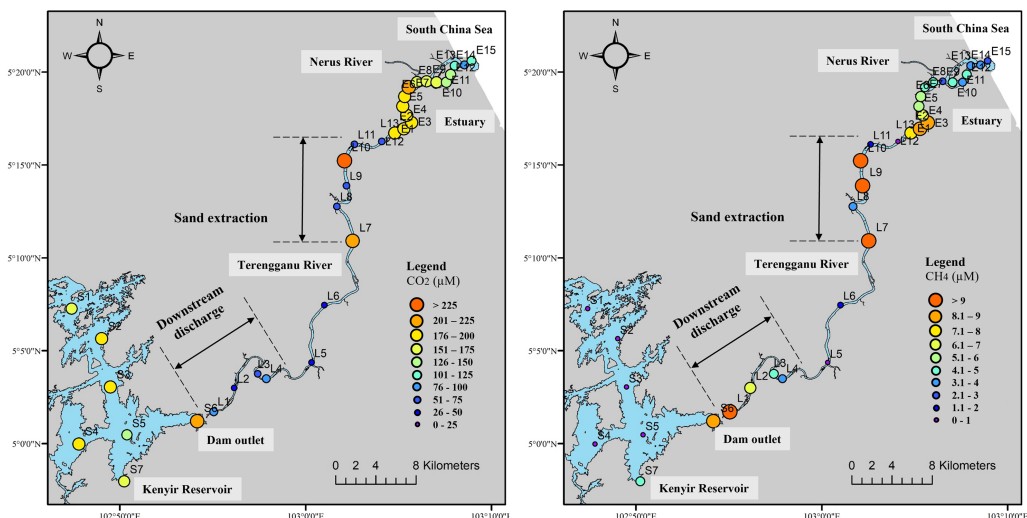

**Figure 6** **Greenhouse gases concentration in the entire Terengganu River system.**

dam discharge outlets (L1–L3), in river sections affected by sand mining (L7–L9), and in the upper estuary receiving urban drainage (E1–E4). Elevated $CO_2$ levels were detected in the Kenyir Reservoir and upper estuary, particularly near sand extraction sites (L7) and at urban tributary outlets (E6).

This study highlights the critical role of hydropower reservoirs, such as Kenyir dam, as significant $CO_2$ and $CH_4$ emissions sources. The findings reveal substantial temporal and spatial variability in GHG emissions within a tropical river catchment, emphasizing how human activities, including river damming, sand extraction and uncontrolled urban drainage discharge, impact the river's GHG dynamics. These insights underscore the need for further research to assess the long-term impacts of dam operations, urbanization, and other anthropogenic influences on GHG emissions in the Terengganu River system.

## ACKNOWLEDGEMENTS

We sincerely thank the Centre of Research & Field Service, Universiti Malaysia Terengganu, for providing assistance with the sampling boat in Kenyir Reservoir, as well as in the Terengganu River and estuary.

### Funding

This work was supported by the Malaysia Ministry of Higher Education-Fundamental Research Grant Scheme (FRGS/1/2015/WAB05/UMT/02/1). The funders had no role in study design, data collection and analysis, decision to publish, or preparation of the manuscript.

## Grant Disclosures

The following grant information was disclosed by the authors:

Malaysia Ministry of Higher Education-Fundamental Research Grant Scheme: FRGS/1/2015/WAB05/UMT/02/1.

## Competing Interests

The authors declare there are no competing interests.

## Author Contributions

- Daryl Jia Jun Lee conceived and designed the experiments, performed the experiments, analyzed the data, prepared figures and/or tables, authored or reviewed drafts of the article, and approved the final draft.
- Siti Farhain Mohd Ludin conceived and designed the experiments, performed the experiments, analyzed the data, prepared figures and/or tables, authored or reviewed drafts of the article, and approved the final draft.
- Wei Wen Wong conceived and designed the experiments, analyzed the data, prepared figures and/or tables, authored or reviewed drafts of the article, and approved the final draft.
- Liyang Zhan conceived and designed the experiments, analyzed the data, authored or reviewed drafts of the article, and approved the final draft.
- Seng Chee Poh conceived and designed the experiments, performed the experiments, analyzed the data, prepared figures and/or tables, authored or reviewed drafts of the article, and approved the final draft.

## Data Availability

Raw data is available in the Supplemental Files.

## Supplemental Information

Supplemental information for this article can be found online at http://dx.doi.org/10.7717/peerj.19929#supplemental-information.

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
