# Peer review of "An assessment of CO2 and CH4 emissions in a tropical river: from the Kenyir Reservoir to the estuary"

_PeerJ, doi:10.7717/peerj.19929_

## Round 0.1 · original submission · Major Revisions

A substantial data set is presented to carry out a comprehensive study of the CO2 and CH4 emissions in a reservoir system. However, the manuscript needs to be restructured and several sections require to be clarified, as indicated by the reviewers. Moreover, conclusions should be supported by a more profound statistical analysis.

·

Basic reporting

The article contains an interesting dataset of some rivers and reservoirs not well covered, and is a very relevant set of observations.
The text requires some structural changes:
1. The Abstract has no context, and the reporting of results is too detailed.
2- The Discussion is full of results, which means that they should move to the results and provide better context in the discussion.
3. Many citations are either missing in the reference list, or are not relevant to the sentence where they are. see "other comments" for specific examples
4. The methods lacks some important aspects, for example how is k estimated for the river and how the upscaling for the whole system is performed.

Experimental design

The design is correct, even though the methods require some details as I mentioned. Particularly how is ebullition estimated and separated from diffusive fluxes.
The research question and knowledge gap is a bit vague, but ok given the descriptive nature of the study

Validity of the findings

In general, some statements with causal links are not well supported with the results, so the whole discussion needs to be revisited, removing the results pieces and threading better how it links with the conclusions.

Additional comments

L24: The abstract needs a couple sentences introducing the topic and ewhy is necessary to study spational and seasonal varations in GHG.
L33: The result description is too detailed for an abstract, for instance explainign what are the sppcific patterns single site sis not necessary in abstracts.(for instance those details about site 7).
L40: Not sure that the causal link between sand extraction and GHG emissions is that well stablished for this sentence. I would remove this unless you have a clear evidence.
L65: This needs some citations on how dam construction affects GHG emissions.
L87: typo in undersaturated.
L98: Greenhouse gas here is not abbreviated, be consistent throughout the text.
L133: When were these campaigns carried out?
L186: It is a bit unclear how the fluxes measured with this chambers are only ebullition and do not capture also diffusive fluxes.
L229: The main methods needs some more detail on how k was estimated, and not only refer to the SM. For example the SM has only equations for laks and estuaries. How was k in the river section estimated?

L257: Why are equation numbers going from vii to x?
L258: There is no information on how the upscaling was performed.

L262: Avoid sentences saying: Figure xx shows that .... . Rather say: "The reservoir was thermally stratified during both dry and wet periods (Figure S2)". This occurs throughout the text so revise along it, not just here.
L263: I don't undertsand why oxygen saturation can be negative in Figure S2.

L336: How can CH4 concentrations be negative? double check the values.

L349: Multiple part sof the discussion are results, they should move to that section. The first two sentences here, but also in 362, 372...throughout the discussion
L376: Freixa et al 2017, has no methane data. Why is it cited here?. This whole paragraph needs to be reassessed, with results to the results section, and the context with citations much more clearly linked.
L422 Cannot find the conrad citation in the reference list.
L490: typo in reported

Reviewer 2 ·

Basic reporting

This work by Jun Lee and collaborators on CH₄ and CO₂ emissions from a comprehensive reservoir system—comprising a reservoir, a downstream river, and an estuary—is interesting, novel, and highly meritorious. In many cases, greenhouse gas emissions are reported for isolated segments of an ecosystem, and it is rare to see studies that address an entire system, as is done here. Overall, this work, supported by a substantial data set, deserves to be published. However, in its current form, the manuscript falls short in several important aspects, which is why I strongly encourage the authors to undertake a thorough and comprehensive revision of their valuable work.

Positive aspects:
Comprehensive study of a complete reservoir system.
Conducted over multiple dates.
Extensive number of sampling stations.
Detailed and exhaustive incorporation of literature data.

Negative aspects:
Lack of clarity in several sections, as detailed below.
Very limited statistical analysis.
Some conclusions are arguable.

Other basic comments:

The language is perfectly understandable but would benefit from a polishing step to improve overall readability.

The abstract feels very weak and may give the reader an incorrect impression of the study. I recommend telling a brief, cohesive story rather than simply copying and pasting parts of the results.

The methods are generally sound, but the way the data are treated is unclear. For example, it is not specified whether the sampling dates during the wet and dry seasons were averaged. This could be clarified by enhancing Table S1 with more detailed information.

There is also a noticeable lack of statistical analysis in the work. Only a few mentions of statistical tests are made, beginning at L417, and this issue must be addressed as a priority.

In general, I believe this impressive work would benefit from a thorough revision, with close attention to its completeness, the clear and logical flow of ideas, and distinguishing between hypothetical/arguable statements and demonstrated claims.

Experimental design

The experimental design is sound but presents several gaps that need to be addressed. In particular, I would suggest the following:

Include a clear statistical analysis.

The method used to generate heatmaps should be included. If spatial interpolation techniques like kriging or inverse distance weighting (IDW) were used, they must be described in detail, along with any assumptions made during the analysis.

Provide more detail on the sampling strategy and explain how the obtained results were integrated into the key findings. For example, when the authors mention “during the wet season, results...,” it is unclear whether the presented results are the average of all measurements across all stations.

In some instances, excessive detail is included in the methods section, such as the calculation of water density (Eq. vii). Conversely, in other cases, insufficient detail is provided, or the information is contradictory. An example of this is the wind speed consideration: while it is stated that wind speed was factored in, only n = -1/2 was considered (Eq. ii and L220).

See additional examples in Section 4 of this review.

Validity of the findings

In general, the findings are valid, and some are even quite remarkable. I am referring specifically to the heatmaps and Figure 5. The latter represents a particularly important input in this work.

However, I also feel that the Figures are not optimally presented. It seems they were prepared by different authors and then grouped together, sometimes inefficiently. I would suggest creating new figures with more attention to detail. For instance, Figure 1 includes several sub-figures that could be reorganized more efficiently. Figure 2 is overly large and contains duplicate information; I suggest a clearer presentation of the wet/dry seasons. Figure 3 could be compacted, and Figure 4 (heatmaps of fluxes) should use a consistent color scale for easier comparison. Additionally, Figure S2 is of very poor quality and is excessively large.

Some findings are also somewhat arbitrary and questionable. For example, the claim that the reservoir is monomictic seems debatable, as clear stratification is observed in both seasons (Figure S2). Further similar observations are presented in Section 4 of this report.

Additional comments

Please find additional comments below, some of which are major.

Figure 1: I would suggest compacting this figure, which should be straightforward, to allow for better integration within the printed article.

Table S1: The information provided is not precise enough. For example, depth profiles are mentioned for the estuarine system (L150) but are not described in Table S1.

Study Site and Sampling: Please provide more information, such as the mean depth of the water bodies. It is unclear why river sampling was classified as high and low discharge when these do not seem to correspond with dry and wet periods. For instance, the same month is classified as high discharge in 2018 and low discharge in 2019. Additionally, are the five reservoir samplings averaged per period?

Figure S2: The figure is unclear; I had to paste it into a Word document to see the graph details. The Kenyir reservoir graphs do not indicate the station or date of measurement. What does SD stand for in these graphs? Please also indicate the dry/wet periods in the estuary heatmaps. Why are some areas blank in the heatmaps?

Heatmaps: How were the heatmaps generated? Please clarify the methodology used.

Figure 4: I recommend using the same color scale across the flux maps to facilitate comparisons.

Statistics: There is a general lack of statistical analysis in the manuscript. This needs to be addressed in the relevant sections.

Results: No results on ebullition are presented, even though they are mentioned in the discussion (L372-379). These results should be moved to the Results section.

L352: The claim that Kenyir is a monomictic reservoir seems contradictory based on the observed stratification. Please clarify.

L354-356: There is no evidence to support this claim.

Figure 2: "February 2018" – I suggest indicating which dates correspond to the wet/dry periods.

L357-360: Is there any evidence that wind conditions were higher during the wet season?

L384-386: Wind data should be presented. How does turbulence within the water column enhance redissolution or rising bubbles?

L401: "The higher greenhouse gas emissions in tropical reservoirs…" – Higher compared to what? Please clarify the comparison.

L417: This is the first mention of a statistical test, but more statistical analysis is needed throughout the manuscript.

In general: Please avoid subjective adjectives like "high." Instead, use precise, quantifiable terms to ensure clarity and accuracy.

L429-430: The writing in this section is unclear and needs improvement.

L442-444: How does an increase in turbidity enhance organic carbon degradation? Please explain the mechanism.

L447: Should be written as "in-stream."

L456: "0.07 µM CH₄" seems incorrect. Based on other data, it should be around 0.002 µM.

L487-489: This section should be toned down, as it presents a hypothesis without clear evidence. Please revise the wording for clarity.

L495-506: I find Figure 5 very interesting, but I do not fully agree with the diel effect explanation for the observed trend. It seems that tidal height was the dominant factor in both cases. I suggest expanding this discussion. This is later touched on in L520-529, so I recommend merging these paragraphs.

L508-518: It seems unnecessary to discuss one negative CH₄ flux in such detail. This result likely indicates that dissolved CH₄ concentrations were below 0.002 µM, close to the detection limit. This is not convincing enough evidence to suggest CH₄ oxidation.

Table 4: Please improve the column headings for clarity.

L556-561: In my view, this section belongs in the Discussion. I also suggest indicating the percentage of total emissions associated with CH₄ and CO₂.

---

## Round 0.2 · Minor Revisions

The manuscript has been revised by the authors and the current version shows clear improvements in relation to the original one. The majority of the issues raised by the reviewers have been addressed appropriately, and the manuscript now presents clearly its results, findings and conclusions.

I would just recommend thorough language revision by a fluent English speaker as the paper still contains many language issues that affect the overall readability and precision of the work.

**Language Note:** The review process has identified that the English language must be improved. PeerJ can provide language editing services - please contact us at [email protected] for pricing (be sure to provide your manuscript number and title). Alternatively, you should make your own arrangements to improve the language quality and provide details in your response letter. – PeerJ Staff

Reviewer 2 ·

Basic reporting

I acknowledge the efforts made by the authors in revising this interesting manuscript. The current version shows clear improvements in both structure and clarity. Several of the issues raised in the first round have been addressed appropriately, and the manuscript now presents its findings in a more coherent and accessible manner. The integration of a substantial dataset covering a reservoir, river, and estuary remains a valuable and distinctive aspect of the study.

I also recognize the authors' effort to improve the manuscript’s language, as indicated in their response and reflected in some sections of the revised text. However, despite this proofreading step, the manuscript still contains numerous spelling errors, awkward phrasing, and unclear constructions. These language issues are frequent enough to affect the overall readability and precision of the manuscript. While the scientific content remains generally understandable, I strongly recommend a more thorough language revision, ideally with the assistance of a fluent English speaker or professional editing service.

The clarity of the Figures has improved in this revised version, and I appreciate the effort made in that direction. However, I encourage the authors to carefully review the captions of all Figures and Tables, as several remain unusually vague, ambiguous, or overly brief to the point of limiting the reader’s understanding. For example, the caption of Figure 3 — “Longitudinal distribution of CO₂ and CH₄ in Terengganu River” — is unclear, as CO₂ and CH₄ are not parameters themselves. It should be specified whether the Figure refers to concentrations, fluxes, or another metric. Furthermore, I suggest avoiding the use of unexplained acronyms in Figure legends, such as “TRE,” to ensure that each figure can be interpreted independently from the main text. Overall, greater attention to detail in the presentation would considerably improve the accessibility and professionalism of the manuscript.

Experimental design

The experimental design is generally well conceived, relying on a set of classical and well-established methods that are appropriate for addressing the study objectives. The combination of approaches is methodologically sound and reflects standard practice in the field.

One potential point of criticism is the determination of dissolved gases used to estimate diffusive fluxes. This was done using the headspace equilibration method from stored water samples, which were preserved without chemical inhibitors. While this raises a concern regarding possible microbial alteration of gas concentrations during storage, I assume that the analyses—conducted at Universiti Malaysia Terengganu—were carried out within a short timeframe, which would help minimize this risk.

Validity of the findings

The findings presented in this manuscript are original, robust, and scientifically valuable. As previously noted, this work by Jun Lee and collaborators—focusing on CH₄ and CO₂ emissions from a comprehensive reservoir system that includes a reservoir, downstream river, and estuary—is both novel and commendable. While most studies report greenhouse gas emissions from isolated segments, this manuscript offers a more integrated view of an entire aquatic continuum, which remains rare in the literature. The current version of the manuscript is supported by a substantial dataset and employs a set of well-established methods, lending credibility to the results. The conclusions are generally well supported by the evidence provided, and the study offers important insights into greenhouse gas dynamics in tropical systems. Despite some remaining issues, particularly in language clarity and presentation, the scientific content is solid and merits publication after appropriate revisions.

Additional comments

Minor Comments

Please note that to identify clearly page and line numbers, I used as a reference the “Track Changes” version of the Word document.

Abstract

L27: Please define GHG at first use (it is defined on L93 but not in the abstract, where it is used eight times).

In the abstract, I feel that my previous recommendation to distinguish more clearly between hypothetical/arguable statements and demonstrated claims was not taken into account. For instance, L36, L42–43, and L46–47 present three assumptions as if they were demonstrated findings.

L38: It might be useful to mention that emissions from reservoirs usually decrease over time, which may not be obvious to many readers.

L46: Please specify what this “5.5%” refers to.

Materials and Methods

In Study site and sampling subsection: There is no mention of Table S1.

L178: The phrase “monthly discharge rate” is ambiguous, as it is unclear whether it refers to the average flow rate during each month. Since the units are in m³ s⁻¹, it would be more accurate to rephrase as: “The monthly-averaged discharge of the Terengganu River ranged from 147 to 224 m³ s⁻¹.”

Same section: Am I missing something, or is there a discrepancy in the discharge categories? The "monthly discharge rate" is reported to vary from 147 to 224 m³ s⁻¹, but the two conditions considered on L184 are >150 and <120 m³ s⁻¹.

Table S1 is confusing: April is considered both a high- and low-discharge period, and all river sampling was done during the dry season. This is unclear, and I suggest the authors clarify this around L180 with a brief comment.

L202: The sentence is grammatically awkward and unclear. I suggest: “A summary of the study sites and the sampling activities conducted in the reservoir, river, and estuary during each campaign is provided in the Supplementary raw data and Table S1.”

L211–212: These two sentences are somewhat unclear and potentially misleading. The first states that water samples were collected using a Niskin bottle, while the second implies that the 60 mL borosilicate serum bottle was also used for sample collection. Please clarify the procedure.

L233: Figure S1 is missing from the list of supplementary files, while two files are labeled “Figure S3.” One of them includes a legend describing the chamber.

Table S3: It is unclear what the indicated codes refer to (e.g., W92, M95, C&W03…). The legend does not explain the content of the table but instead discusses the choice of a specific model. This discussion should be moved to the main text.

Around L292–307: I still feel that excessive detail is provided for something quite straightforward. Equation viii on L307 might be sufficient. Also, please note that the funnel area (Eq. viii) is referred to as the “wok area” in Eq. v — I recommend using “funnel area” for consistency and clarity.

Results

Figure 2: I did not have access to the full legends of the supplementary Tables and Figures. Please ensure all legends are complete and clear.

L392: Reference is made to Figure 2 (fluxes), while the results presented here are concentrations. Are the concentration results shown in any figure or table?

L394: “CO2 and CH4 fluxes however did not show significant spatial variations (Kruskal-Wallis test, p > 0.05, Fig. 2).” This sentence is too vague — in which section or ecosystem was this observed? I would be surprised if no spatial variation was observed across any stations or seasons.
General: I am uncomfortable with the extensive use of the acronym “TRE” instead of “Terengganu River Estuary.” I believe the acronym is overused. Simply using “Estuary” would often be clear enough once the full name has been introduced in the text.

Discussion

L497: “Oxygen-stratification creates hotspots for CO2 and CH4 production (Fig. S6).” There is no Figure S6. Also, how does oxygen stratification create hotspots for CO2 and CH4 production? This appears to be an oversimplified statement that is not easy for the reader to interpret.

L502: What is the purpose of mentioning ebullition at S7? What is its connection to the previous sentence? Please elaborate.

L505: “Statically”? Should this be “Statistically”?

L505: I do not believe there is sufficient evidence that wind “disrupts stratification.” However, I agree that wind increases superficial water mixing and kᵢ, according to the authors’ own equations.

L520: There is no Figure S6.

L537: “Kenyir Reservoir ranks 9th and 8th for CH4 and CO2 in global freshwater GHG emissions.” This is inaccurate. Kenyir ranks 9th and 8th among the references listed in Table 1, not globally.

L579: “As water flowed from L1 to L4, approximately 80% of the CH4 concentration was reduced (Fig. 3).” The phrasing is poor. Please consider rewording for clarity.

L599: “…severely reducing CO2-fixed microbial community…” — grammatically incorrect; please revise.

Table 2 legend: The phrase “…source apportionment contribution…” is awkward and unclear. Please rephrase.

L738: The Global Warming Potential (GWP) of methane should be mentioned.

---

## Round 0.3 · accepted · Accept

The authors have addressed all of the reviewers suggestions and the manuscript is ready for publication.

Reviewer 2 ·

Basic reporting

I acknowledge and appreciate the significant improvements made by the authors in this revised version of the manuscript. The authors have responded thoroughly to the comments from the previous round of review, and the manuscript is now clearer and better organized. I am particularly pleased to note that the authors have undertaken professional English editing, which has greatly enhanced the readability and overall quality of the text. The writing is now generally clear and precise, and the Figures and Tables captions are more comprehensible. Minor typographical and grammar issues remain but do not substantially affect the manuscript’s readability.

Experimental design

As this is now the third round of review, I have no further comments regarding the experimental design. I consider it sound, appropriate for the study’s objectives, and fully acceptable in its current form.

Validity of the findings

Similarly, I have no further comments regarding the validity of the findings. The results are robust and scientifically significant, and the conclusions are well supported by the data. The manuscript offers valuable insights into greenhouse gas dynamics across a tropical reservoir-river-estuary continuum and contributes meaningfully to the literature.

Additional comments

I believe this study will be well received by the limnology community, as the inclusion of a reservoir, a river, and an estuary provides a perspective that is not often addressed in a single study. This integrated approach strengthens the scientific value of the work and adds an important dimension to our understanding of greenhouse gas dynamics in aquatic systems.